# A serial multiplex immunogold labeling method for identifying peptidergic neurons in connectomes

**Réza Shahidi, Elizabeth A Williams, Markus Conzelmann, Albina Asadulina, Csaba Verasztó, Sanja Jasek, Luis A Bezares-Calderón, Gáspár Jékely\***

Max-Planck-Institute for Developmental Biology, Tübingen, Germany

**Abstract** Electron microscopy-based connectomics aims to comprehensively map synaptic connections in neural tissue. However, current approaches are limited in their capacity to directly assign molecular identities to neurons. Here, we use serial multiplex immunogold labeling (siGOLD) and serial-section transmission electron microscopy (ssTEM) to identify multiple peptidergic neurons in a connectome. The high immunogenicity of neuropeptides and their broad distribution along axons, allowed us to identify distinct neurons by immunolabeling small subsets of sections within larger series. We demonstrate the scalability of siGOLD by using 11 neuropeptide antibodies on a full-body larval ssTEM dataset of the annelid *Platynereis*. We also reconstruct a peptidergic circuitry comprising the sensory nuchal organs, found by siGOLD to express pigment-dispersing factor, a circadian neuropeptide. Our approach enables the direct overlaying of chemical neuromodulatory maps onto synaptic connectomic maps in the study of nervous systems.

## Introduction

A comprehensive understanding of nervous system function requires knowledge of not only precise neuronal connectivity but also the unique combinations of molecules expressed by each neuron. Connectomics using serial-sectioning electron microscopy (EM) aims to map the synapse-level connectivity of entire neural circuits (*Morgan and Lichtman, 2013*). However, wiring diagrams provide an incomplete picture of the nervous system, since they lack information about the molecular nature of neurons.

One important class of molecules are neuromodulators, including monoamines and neuropeptides, that actively shape the output of circuits by modifying synaptic function and neuron excitability (*Bargmann, 2012*; *Bargmann and Marder, 2013*; *Bucher and Marder, 2013*; *Marder, 2012*). The direct mapping of specific neuromodulators to their source cells in synapse-level anatomical maps would help to integrate connectomic and neuromodulatory perspectives, thereby enriching our understanding of circuit function.

Several approaches have been developed to assign molecular identities to neurons on EM sections. Genetically encoded tags with enzymatic activity, such as miniSOG and APEX probes, allow the resolution of cells or subcellular structures by EM following incubation with a substrate that is converted to electron-dense deposits (*Martell et al., 2012*; *Shu et al., 2011*). Fixation-resistant protein tags compatible with EM procedures and observable by correlative light and electron microscopy (CLEM) or directly by immunogold labeling (immunoEM), such as the smGFPs (*Viswanathan et al., 2015*), have also been developed. In CLEM, fluorescently labeled neurons or organelles can be imaged live prior to fixation, sectioning, and EM imaging of the specimen. Fiducial markers allow the correlation of cellular structures in the immunofluorescence (IF) and EM images (*Colombelli et al., 2008*; *Maco et al., 2013*; *Maco et al., 2014*; *Urwyler et al., 2015*). However,

\*For correspondence: gaspar. jekely@tuebingen.mpg.de

**Competing interests:** The authors declare that no competing interests exist.

**eLife digest** In the nervous system, cells called neurons connect to each other to form large "neural" networks. The most powerful method that is currently available for tracing neurons and mapping the connections between them is called electron microscopy. This requires slicing brain tissue into ultrathin sections, which are then imaged one by one. However, while electron microscopy provides highly detailed information about the structure of the connections between neurons, it does not reveal which molecules the neurons use to communicate with each other.

To address this question, Shahidi et al. have developed a new approach called 'siGOLD'. Unlike previous approaches, siGOLD allows signal molecules inside cells to be labeled with protein tags called antibodies without compromising the ability to examine the tissue with electron microscopy. The technique was developed using the larvae of a marine worm called *Platynereis*. A single larva was sliced into 5000 sections thin enough to view under an electron microscope, and 150 of these were selected to represent the entire body. Because neurons are typically long and thin, individual neurons usually spanned multiple slices.

To identify the neurons, Shahidi et al. then applied an antibody that recognizes a specific signal molecule to a subset of the slices. The antibodies were labeled with gold particles, which show up as black dots under the electron microscope. Because the molecules recognized by the antibodies are present all along the neuron, and because individual neurons extend over multiple slices, it was possible to trace single neurons by labeling only a small number of slices. Repeating this process in different subsets of slices with antibodies that bind to different signal molecules allowed entire neural circuits to be mapped.

In the future, Shahidi et al.'s approach could be adapted to study neural networks in other organisms such as flies, fish and mice.

the neuron-specific probes employed by these technologies must be expressed using transgenesis, limiting their use to one or a few markers, such as in the 'two-tag' labeling approach (*Lin et al., 2015*).

An alternative approach to assigning molecular identities to neurons is the use of endogenous targets in fixed specimen, as in array tomography, a method employing sequential IF with several different antibodies on large series of sections (*Collman et al., 2015*; *Micheva and Smith, 2007*). IF imaging can be combined with scanning EM imaging of selected sections to resolve ultrastructural detail. The sequentially acquired IF and EM images can be registered, allowing for correlative analysis of images of different modalities (*Collman et al., 2015*). However, the sequential IF procedure compromises the membrane contrast in EM, preventing the reliable tracing of fine processes across series of sections (*Collman et al., 2015*). Array tomography also relies on embedding the specimen in porous, hydrophilic acrylic resins to allow optimal immunolabeling. For large-scale serial sectioning projects, epoxy resins, such as Epon, are favored due to their higher stability during sectioning and imaging and the optimal ultrastructural preservation they provide (*Bock et al., 2011*; *Briggman et al., 2011*; *Bumbarger et al., 2013*; *Ohyama et al., 2015*; *Randel et al., 2015*; *White et al., 1986*). Unfortunately, the Epon-embedding procedure, including osmium-fixation and resin polymerization at 60°C, compromises the immunogenicity of many endogenous targets (*Brorson, 1998*; *Brorson and Reinholt, 2008*; *De Paul et al., 2012*). Current immunolabeling approaches therefore sacrifice the mechanical stability and ultrastructural contrast of the sample for optimal immunolabeling.

One exception is short amidated neuropeptide antigens that show good immunopreservation in epoxy-embedded samples (*Hamanaka et al., 2010*; *Koizumi et al., 1989*; *Merighi et al., 1992*; *Yasuyama and Meinertzhagen, 2010*). Such antigens represent promising targets in an attempt to combine connectomics on Epon-embedded samples and immunogold labeling for specific neuromodulators. Neuropeptides are often chemically modified, including C-terminal amidation (*Eipper et al., 1992*), which confers stability and high immunogenicity to even very short peptides (*Conzelmann and Jékely, 2012*). Furthermore, neuropeptides show neuron-type specific expression

and are distributed throughout the axon of peptidergic neurons (*Wong et al., 2012*; *Zupanc, 1996*), representing useful markers for neuron-type identification.

Here, we introduce serial-multiplex immunogold (siGOLD), a method for immunolabeling connectomes. siGOLD involves immunogold labeling of small subsets of sections from large series with different neuron-type-specific neuropeptide antibodies. The molecularly identified neurons and their synaptic partners are then reconstructed from the entire aligned series using standard EM-based connectomics.

We established siGOLD using larval stages of *Platynereis dumerilii*, a marine annelid that has recently emerged as a powerful model for circuit neuroscience, genetics, and whole-body connectomics (*Backfisch et al., 2013*; *Bannister et al., 2014*; *Gühmann et al., 2015*; *Randel et al., 2014*; *Randel et al., 2015*; *Tosches et al., 2014*; *Veedin-Rajan et al., 2013*; *Zantke et al., 2014*). We identified several amidated neuropeptide epitopes that showed long-term immunopreservation in Epon-embedded samples, allowing us to simultaneously obtain high ultrastructural detail and specific immunogold signal. Using siGOLD with 11 distinct antibodies on the same specimen, we identified several neuropeptide-containing neuron profiles in a whole-body *Platynereis* larval serial EM dataset (*Randel et al., 2015*). Furthermore, taking advantage of the whole-body series, we fully reconstructed several peptidergic neurons identified by siGOLD in the *Platynereis* larva. We also identified and reconstructed the postsynaptic partners of selected peptidergic neurons, focusing on the nuchal organs, paired, putatively chemosensory organs with high structural complexity and variability among the annelids (*Purschke, 1997*; *Purschke, 2005*; *Purschke et al., 1997*; *Schlötzer-Schrehardt, 1987*). Our work demonstrates that siGOLD can be used in large serial EM datasets to assign molecular identities to multiple neurons using different markers and to fully reconstruct and analyze the synaptic connectivity of these neurons at EM resolution.

## Results

### Multiplex neuron identification with siGOLD on serial sections

In order to selectively label individual neurons in large-scale serial EM datasets, we established an immunoEM procedure to label ultrathin sections with neuronal cell-type specific antibodies. We reasoned that immunoEM performed on only a few layers from a large series of sections could identify neuron profiles that contain the antigen (*Figure 1A*). We first performed immunoEM on 40-nm serial sections from the ventral nerve cord (VNC) of a 72 hr post-fertilization (hpf) *Platynereis* larva (specimen HT9-5, *Figure 1B,C*). For specimen preparation, we used a conventional serial TEM protocol including high-pressure freezing, fixation with a freeze substitution medium containing 2% osmium tetroxide and 0.5% uranyl acetate, and embedding in Epon. We also developed a procedure for the safe handling of several grids in parallel during the immunostaining and contrasting procedure. We optimized the immunolabeling protocol to achieve high specificity for immunoEM and high ultrastructural detail. In our protocol, we use secondary antibodies coupled to ultra small gold particles and a silver-enhancement procedure. We also fine-tuned the contrast-staining protocol to optimize contrast for both gold labeling and ultrastructural detail.

In preliminary tests, we found strong and localized labeling in neurites using 11 different polyclonal antibodies generated against short amidated neuropeptides of *Platynereis* (*Table 1*).

To test the specificity and reproducibility of immunoEM with the 11 neuropeptide-antibodies, we collected two sets of transverse serial sections from the first trunk segment of the HT9-5 specimen. Four to 18 consecutive serial sections were collected on each EM grid for immunoEM. For each antibody, we stained two grids separated by a serial distance of approximately 50 sections (*Figure 1C*). We imaged the VNC region in each section at a resolution of 2.22 nm/pixel followed by stitching and alignment of the images. We found strong and localized labeling in only a small subset of neurites for each antibody (*Figure 2A*, *Shahidi, et al., 2015*). In consecutive sections, the same neurite was often strongly labeled with the same antibody (*Figure 2A*, FVa, layers 1–5). In sections collected on different grids that were labeled with different antibodies, we found distinct patterns of neurite-specific labeling (*Figure 2B,C*). In many sections, we could observe dense core vesicles (DCVs) in the cytoplasm of the gold-labeled neurites, indicative of the peptidergic nature of these cells (*Figure 2A–F*, *Shahidi, et al., 2015*). In high-resolution (0.22 nm/pixel) images, we could observe gold labeling associated with DCVs (*Figure 2D–F*), suggesting that our immunoEM procedure

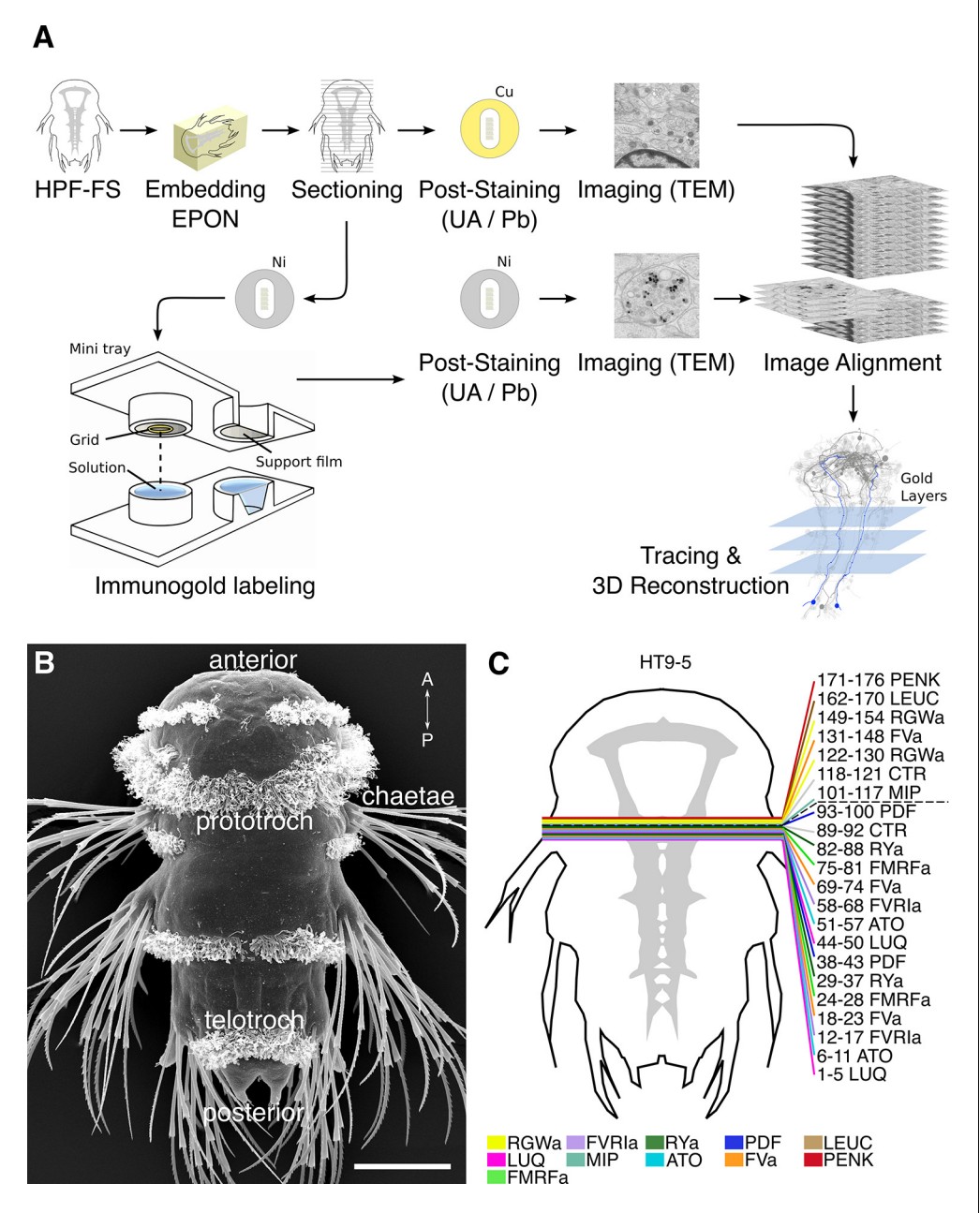

**Figure 1.** Development of the siGOLD method. (**A**) Schematic flowchart of the siGOLD labeling approach from high-pressure freezing and freeze substitution (HPF-FS) to tracing and 3D reconstruction. Ni, nickel grid, Cu, copper grid. (**B**) SEM micrograph of a 72 hpf *Platynereis* larva. (**C**) Schematic of the HT9-5 sample showing the position of the ventral nerve cord (VNC), ventral view. Colored lines indicate where cross-sections through the VNC were taken, near the base of the circumesophageal connectives at the level of the first commissure. Layer number(s) followed by neuropeptide ID are indicated for each colored line. Dashed line indicates the gap (approximately 10 missing sections) between the first and second series of sections. Scale bar: (**B**) 50 µm.

labeled mature neuropeptides residing inside these vesicles. Several ultra-small gold particles were not enlarged during the silver-enhancement procedure but were also associated with DCVs (*Figure 2D–F*).

Next, we comprehensively scored the distribution of gold particles in the VNC series (*Figure 3*). We selected all neurites that were labeled with two or more gold particles in any of the immunoEM

**Table 1.** List of antibodies used

| NP precursor name | Abbreviation | Antigen |
| --- | --- | --- |
| FMRFamide | FMRFa | (C)FMRFa |
| RYamide | RYa | (C)VFRYa |
| Myoinhibitory peptide/Allatostatin B | MIP | (C)AWNKNSMRVWa or (C)VWa |
| RGWamide | RGWa | (C)RGWa or (C)GWa |
| Proenkephalin | PENK | (C)YGDLSFSNSNYa |
| Luqin | LUQ | (C)WRPQGRFa |
| Allatotropin | ATO | (C)GFRTGAYDRFSHGFa |
| Pigment dispersing factor | PDF | (C)NPGTLDAVLDMPDLMSLa |
| Leucokinin | LEUC | (C)KFTPWAa |
| FVamide | FVa | (C)AHRFVa or (C)FVa |
| FVRIamide | FVRIa | (C)FVRIa |

The full name and abbreviation of neuropeptide precursors that contain the neuropeptides used for immunoEM. The FMRFa, RYa, MIP short and long, RGWa, FVa short and long, and FVRIa antibodies have been described previously (**Conzelmann and Jékely, 2012**; **Conzelmann et al., 2011**; **Conzelmann et al., 2013a**; **Jékely et al., 2008**). All 11 neuropeptides are amidated (a). A Cys (C) was added to the N-terminus of each peptide to allow coupling during immunization and affinity purification. All antibodies were generated in rabbits. For FVa long, a rat antibody was also generated.

sections and traced these neurites across all sections. We also selected 50 control neurites along a coronal transect spanning the VNC and traced these across all sections (**Figure 3A**). We then counted all gold particles in each traced neurite and tabulated the results (**Figure 3—source data 1**). The gold particles were also summed for each antibody to show a concise summary of all gold labels (**Figure 3B and D**). Since not all ultra-small gold particles were enhanced during silver-enhancement (**Figure 2D–F**) and non-enhanced particles are not visible on the lower resolution images we used for scoring, the gold counts likely underestimate the intensity of gold labeling.

Our quantifications revealed a high neurite-specificity of immunogold labeling with all 11 neuropeptide antibodies (**Figure 3**). In sections where we omitted the primary antibody, we did not see any gold labeling in any of the traced neurites (**Figure 3B**). The VNC in our sample contained approximately 1600 neurite cross-sections and the majority of these showed no gold labeling. However, for each antibody we could identify a small number of neurites that consistently showed strong labeling across different sections (**Figure 3**, **Figure 3—source data 1**). The pattern of the labeled neurites in the VNC showed bilateral symmetry, supporting the labeling of specific peptidergic neuron populations on the left and right sides of the body. Most neurites that were labeled on one grid were also strongly labeled on their grid pair with the same antibody separated by approximately 50 sections. There were some exceptions, for example neurite n5 that was labeled strongly with the FVamide antibody in layers 18–23 but not in layers 69–74 (**Figure 3—source data 1**). To determine whether the lack of labeling was due to the lack of DCVs in these sections, we scored the number of DCVs across all sections for selected neurites (**Figure 3—figure supplement 1**). We found that DCVs were non-uniformly distributed in the neurites and some sections completely lacked DCVs. Importantly, we only detected gold labeling on sections that contained DCVs in the respective neurite (**Figure 3—source data 1**).

siGOLD also allowed us to detect the coexpression of some neuropeptides in the same neurons. For example, we found coexpression of FVa and PDF neuropeptides in a subset of the neurites labeled by these two antibodies (**Figure 3D**). Some antibodies showed extensive overlap in labeling, including the FMRFa, luqin and RYa antibodies.

To test whether the antibodies specifically recognize the neuropeptides used for immunization, we performed morpholino-mediated knockdown experiments of proneuropeptide expression. The specificity of the MIP antibody was demonstrated previously (**Williams et al., 2015**). We performed microinjections with 10 different translation-blocking morpholinos, one targeting each remaining

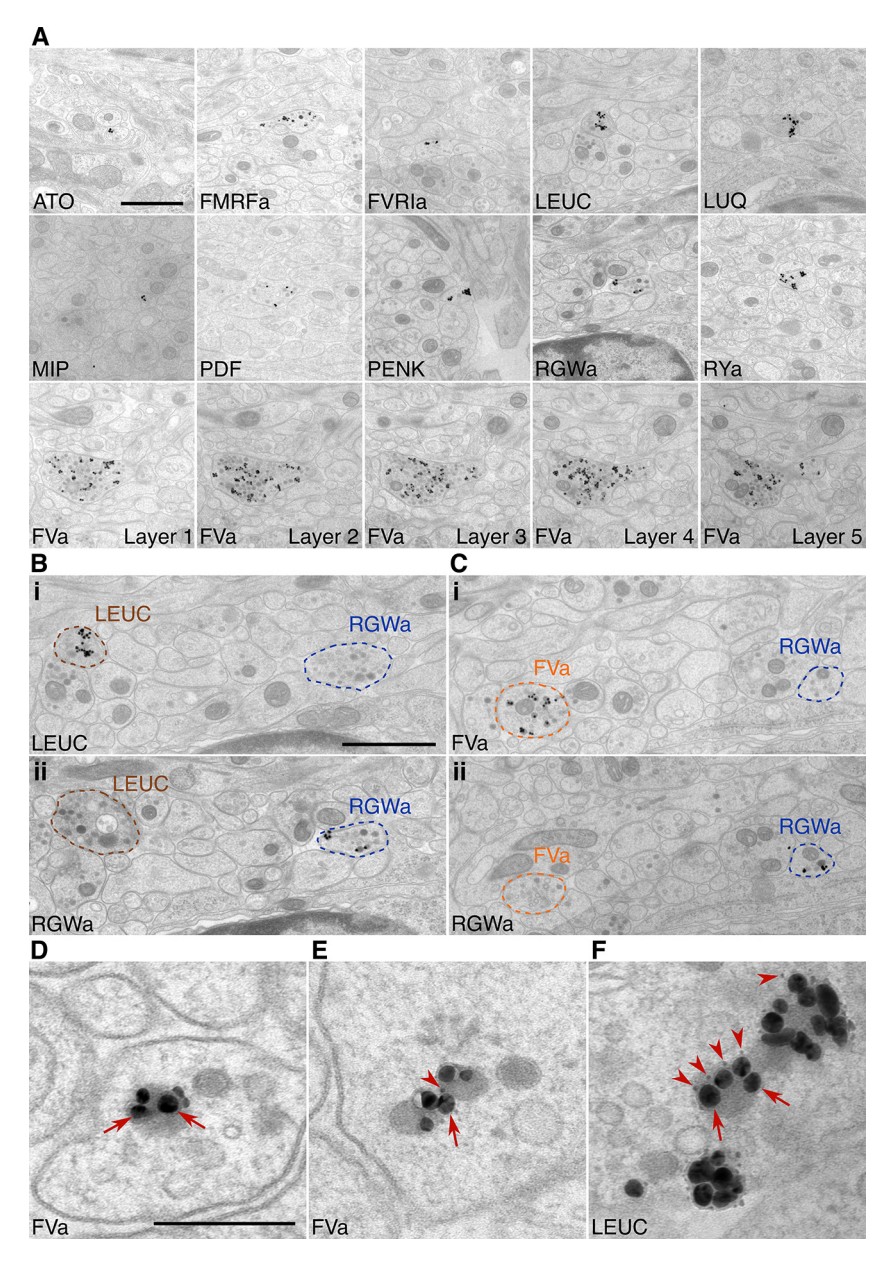

**Figure 2.** Immunolabeling with neuropeptide antibodes on Epon sections. (**A**) Representative micrographs with immunogold labeled axons for the neuropeptide antibodies indicated. For the FVa neuropeptide antibody, five adjacent sections are shown (FVa Layer 1–5). (**B, C**) Neurite-specific labeling in adjacent sections (seven sections apart) labeled with different antibodies. (**D–F**) High-resolution micrographs of immunogold labeled, silver-enhanced gold particles (arrows), and unenhanced ultra small gold particles (arrowheads) Scale bar: (**A-C**) 1 µm; (**D–F**) 200 nm. High-resolution images are available in (*Shahidi, et al., 2015*).

proneuropeptide. We then performed triple IF in whole-mount 72 hpf larval samples using an acetylated tubulin antibody, a rat FVa antibody, and the respective rabbit neuropeptide antibodies. The acetylated tubulin antibody allowed us to exclude developmental abnormalities caused by morpholino injection. The rat FVa antibody provided a further control to exclude that morpholino injection affected proneuropeptide processing in general. For 8 out of 10 antibodies (FVa, FMRFa, ATO, FVRIa, PDF, RGWa, PENK, LEUC), morpholino injection strongly reduced IF signal with the respective antibody (*Figure 4—figure supplement 1*). For the FMRFa antibody, we did not detect staining

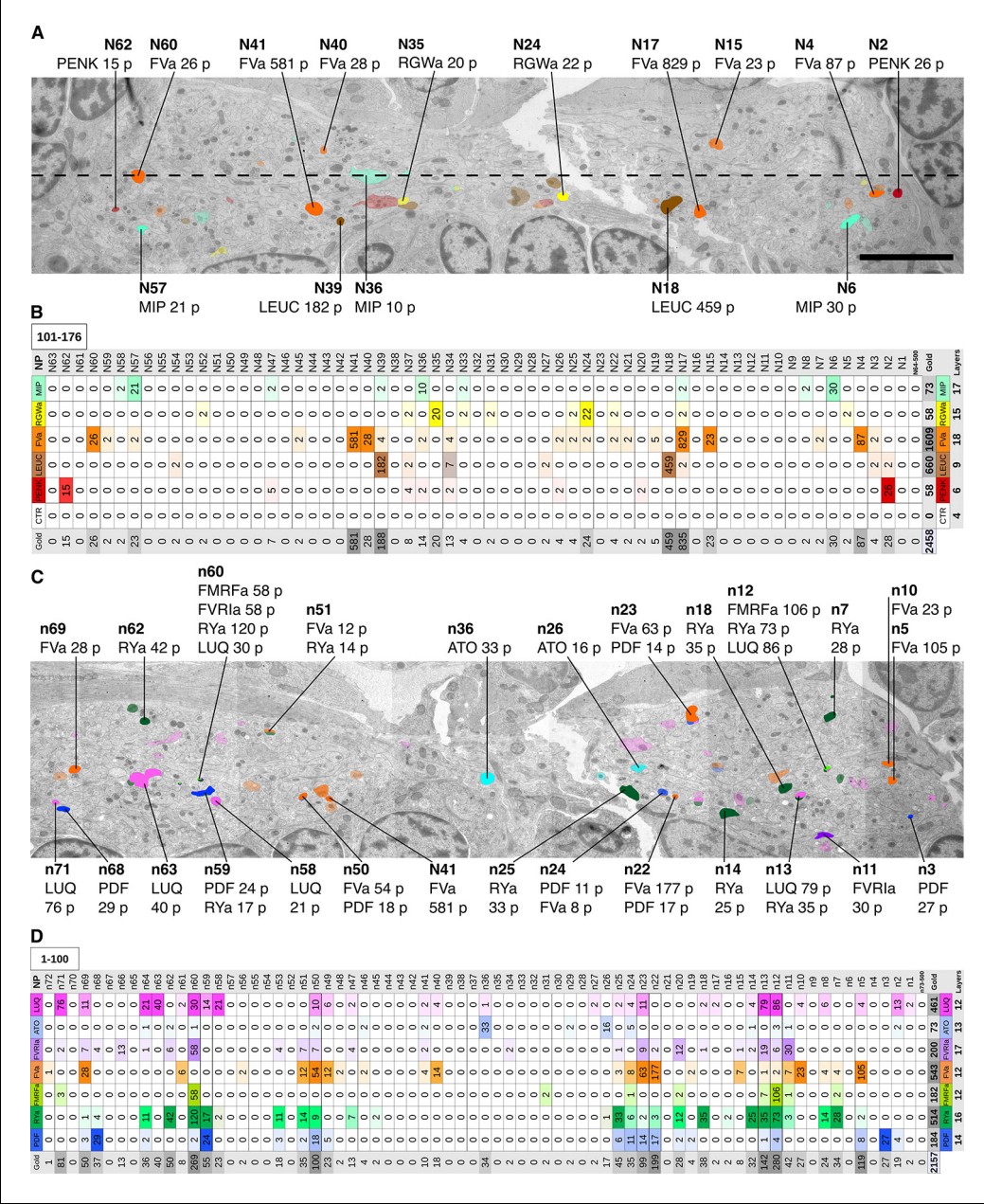

**Figure 3.** Identification of peptidergic neurites by siGOLD in a 72 hpf *Platynereis* specimen (HT9-5). (**A, C**) Anterior view of EM cross-section through the VNC near the first commissure. Dorsal side of larva is to the top. Strongly labeled neurites were analyzed across the whole VNC region. Control axon profiles were analyzed along a transect (dotted line), two axon profiles were sampled every 1 μm. Total of 72 and 63 axons were examined for the first and second series of sections respectively (an approximately 10-section gap occurs between the two series). Colored cell profiles indicate gold labeled neuropeptidergic axons. Different shades of a color represent an approximation of labeling intensity. Positive axons are tagged with neuron number, neuropeptide name, and total number of gold particles per total number of layers for that neuropeptide. (**B, D**) Tables show number of gold particles per axon for each neuropeptide. All strongly labeled axons across the VNC and control axons along the sampled transect are shown. Data were arranged according to the spatial distribution of the corresponding neurites in the VNC. Each sampled axon was traced across all layers and counted for its total number of gold particles. Columns indicate neurons and an ID with 'n' is given to each neuron in the first series of sections and 'N' for neurons in the second series of sections. Rows indicate neuropeptide immunogold labels. Different shades of the same color indicate intensity of gold labeling. Totals are shown for each row (neuropeptide) and each column (axon profile). In

*Figure 3 continued on next page*

*Figure 3 continued*
the final column of each table, the total number of layers stained for each neuropeptide is shown. Scale bar: 5 μm. Gold scores are available in *Figure 3—source data 1*.
The following source data and figure supplement are available for figure 3:

**Source data 1.** Tabulated counts of all gold particles in each traced neurite in HT9-5.
**Figure supplement 1.** Number of DCVs in selected neurite profiles along 100 layers.

in the VNC following morpholino injection but we could still detect staining in the head. The FMRFa antibody therefore likely recognizes other antigens, such as other RFa peptides in the head (*Conzelmann et al., 2013b*). For the RYa and LUQ antibodies, we did not see a reduction in staining intensity in the VNC following morpholino injection. These two antibodies may cross-react with other R[Y F]amide peptides as suggested by the labeling of an overlapping set of neurites with the FMRFa, LUQ, and RYa antibodies (*Figure 3*). Without knowing the exact antigen specificity for these antibodies, we can nevertheless use them as specific markers of R[Y|F]amidergic neurons.

## Comparison of siGOLD labeling to whole-mount immunofluorescence

siGOLD labeling of consecutive sections with 11 antibodies revealed the arrangement of several peptidergic neurites in the VNC (*Figure 3*). To test if this spatial arrangement is consistent between siGOLD and IF labeling of whole specimens, we analyzed the labeling with all 11 antibodies in whole-mount *Platynereis* 72 hpf larval samples. All antibodies labeled subsets of longitudinal axons spanning the entire length of the VNC and occurring in different mediolateral positions (*Figure 4*).

To display the spatial relationships of 11 distinct IF labels, we employed image registration to a reference template of whole-body confocal scans. High-accuracy image registration of different specimens is possible in *Platynereis*, due to the stereotypic anatomy and neuronal connectivity of the larvae (*Asadulina et al., 2012*; *Randel et al., 2015*; *Tomer et al., 2010*). For optimal VNC registration, we generated an unbiased average whole-body reference template based on the non-rigid registration of the acetylated tubulin IF signal from 36 specimens. We aligned whole-body confocal scans of IF specimens to this reference using non-rigid image registration (*Figure 5A,B*; *Video 1*). We then took virtual cross sections from this registered dataset to analyze the spatial relationships of distinct peptidergic axons (*Figure 5C–E*). The arrangement of peptidergic axons in the VNC reconstructed by IF and image registration was similar to that of the VNC reconstructed by siGOLD (*Figure 5F–H*). To allow a more direct comparison of the spatial relationship of peptidergic axons, we performed double IF with each of the RGWa, PDF, and LEUC antibodies generated in rabbits in combination with a FVa antibody generated in rats. The spatial relationships of the neurites labeled with the RGWa, PDF, and LEUC antibodies relative to six prominent FVa neurites was very similar between IF and siGOLD (*Figure 5I,J*). We could also detect the coexpression of FVa and PDF in some axons, both by double-IF and siGOLD. The high specificity of immunoEM, the reproducibility of labeling across sections, and the spatial correspondence of the labels between siGOLD and whole-mount IF indicate that we could accurately identify different peptidergic neurites in a serial EM dataset.

## Application of siGOLD to a whole-body connectome dataset

Next, we tested siGOLD on a whole-body EM series of 5056 sections encompassing an entire 72 hpf *Platynereis* larva (specimen HT9-4) (*Randel et al., 2015*). During sectioning of this larva, we set aside series of 1–6 sections for later immunoEM (*Figure 6—source data 1*).

In the full-body series, we used the 11 rabbit antibodies (*Table 1*) to label a total of 154 sections (3% of all sections) distributed along the length of the larva (*Figure 6A*). We used at least two grids for each antibody, separated by up to 1000 sections. Gold labeling in a whole-body context allowed us to identify several strongly labeled neurite profiles throughout the body. It is worth noting that sections were successfully gold labeled up to 3 years after sectioning the specimen, demonstrating the long-term stability of neuropeptide antigens in Epon sections.

We identified and traced 83 neurons (67 of them with a soma; *Figure 6B*; *Video 2*) with different peptidergic identities. We mapped the distribution of peptidergic axons crossing a section in the first trunk segment in a position comparable to that of HT9-5 and found similar patterns across both siGOLD-labeled larvae and the IF samples (*Figure 7A,B*; compare to *Figure 5*). For selected neurons, we quantified the number of gold particles in every immunolabeled section in HT9-4 (*Figure 8*, *Figure 9*). These counts again demonstrated that we could repeatedly label the same neurons in different sections, often spaced several hundred sections apart.

The reconstruction of several neurons in the whole-body serial EM dataset allowed us to further test the specificity of the immunogold labels by comparing the morphology and position of reconstructed neurons to neurons that were identified by whole-body IF. We found comparable cellular morphologies and positions for six FVa, several PDF (*Figure 8*), two RGWa, and four MIP neurons (*Figure 9*) between siGOLD and IF. This close anatomical correspondence further supports the specificity of siGOLD and demonstrates that the method can be used to unambiguously tag fully reconstructed neurons in serial EM.

## Connectome reconstruction of a siGOLD-labeled peptidergic circuit

The whole-body serial EM dataset combined with siGOLD labeling allows the reconstruction of circuits of neurons with specific peptidergic identities. To demonstrate this, we focused on the nuchal organs in the *Platynereis* larval head as an example. The nuchal organ is a paired putative chemosensory organ in the annelid head with sensory neurons projecting a sensory dendrite into an olfactory pit. The olfactory pit is covered by cuticle and is associated with a patch of motile cilia (*Figure 10A–*

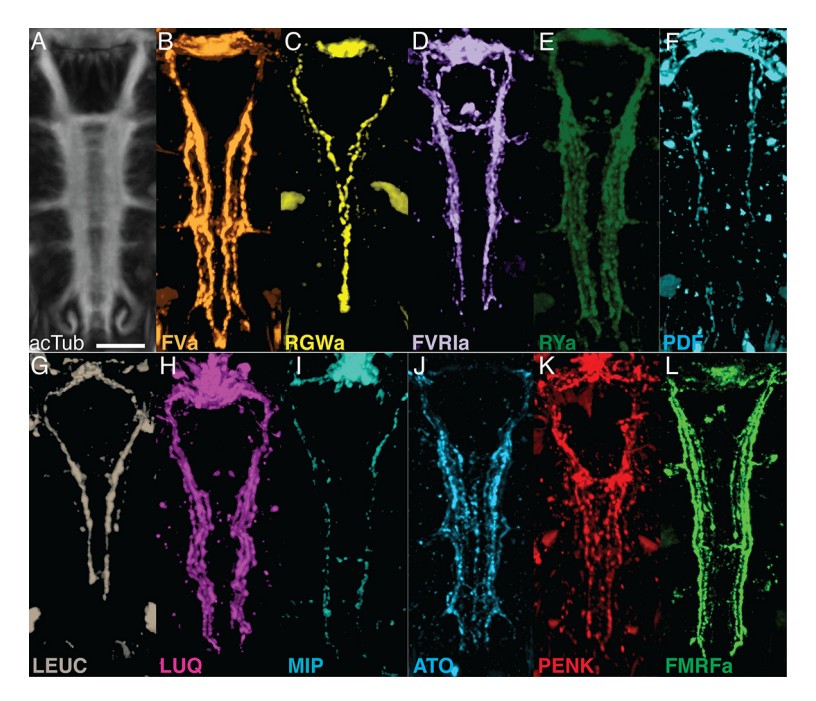

**Figure 4.** Whole-mount IF of *Platynereis* larvae with antibodies raised against neuropeptides labels distinct subsets of neuronal tracks in the VNC. (**A**) Ventral view of *Platynereis* VNC, stained with an anti-acetylated tubulin antibody. This is the registered average VNC generated from scans of 36 larvae. (**B–L**) Whole-mount IF of *Platynereis* larvae with an antibody raised against (**B**) FVa, (**C**) RGWa, (**D**) FVRIa, (**E**) RYa, (**F**) PDF, (**G**) LEUC, (**H**) LUQ, (**I**) MIP, (**J**) ATO, (**K**) PENK, (**L**) FMRFa. Whole-mount scans were cropped to show only the VNC region. Scale bar: 20 μm.

The following figure supplement is available for figure 4:

**Figure supplement 1.** Morpholino-mediated knockdown of proneuropeptides followed by whole-mount IF indicates antibody specificities.

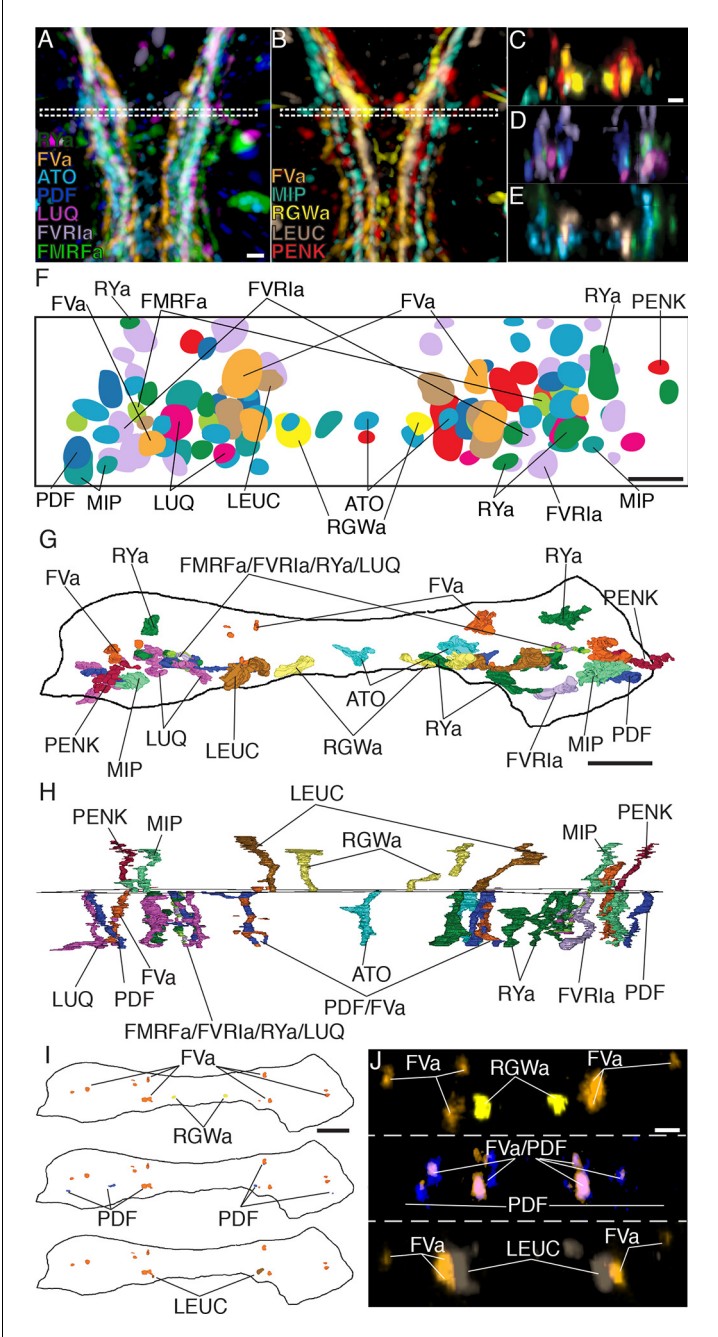

**Figure 5.** Axonal arrangements in the VNC detected by whole-body IF spatially match those detected by siGOLD. (A, B) Ventral overview of individual registered full-body IF with antibodies raised against 11 different neuropeptides (colors). Image is cropped to show only the VNC in the first segment. White dashed box indicates the region where a 5 μm virtual transverse section shown in (C–E) was taken. (C–E) 5 μm virtual transverse section of individual registered full-body IFs from (A, B) in anterior view to indicate their spatial position in the VNC relative to each other. (C) RGWa, FVa, PENK, MIP, (D) FVRIa, PDF, LUQ, FMRFa (E) RYa, LEUC, ATO. (F) Schematic overview of (C–E) indicating relative positioning of individual registered antibody stainings in the VNC, anterior view. (G, H) Reconstruction of neurites labeled by siGOLD in the VNC of specimen HT9-5 with antibodies raised against 11 different neuropeptides, (G) anterior view, (H) ventral view. For comparison with registered IF labeling in (A–F). (I) Position of neurites in specimen HT9-5 siGOLD-labeled with FVa and RGWa (top), FVa and PDF (middle), and FVa and LEUC antibodies (bottom). For comparison with double-IF in (J). (J) 2 μm virtual transverse sections of the VNC of 72 hpf *Platynereis* larvae double-stained with the FVa antibody (orange) and the RGWa (yellow, top), the PDF (blue, middle) or the LEUC (brown, bottom) antibodies. Scale bars: (A–E) 15 μm, (F–J) 5 μm.
*Figure 5 continued on next page*

*Figure 5 continued*

The following source data is available for figure 5:

**Source data 1.** Acetylated tubulin reference signal used for image registration.

*C*; *Video 3*) (*Purschke, 2005*). We found that several sensory neurons of the nuchal organs (SN[nuch]) were strongly labeled by the PDF antibody in three different PDF-labeled sections in the head (*Figure 6A*; *Table 2*). Of the 35 SN[nuch] cells we identified, 15 were labeled with the PDF antibody, with some of the neurons showing strong gold labeling in multiple sections (*Table 2*). This is consistent with the labeling of the nuchal organ with the PDF antibody in IF (*Figure 8G*). We traced all SN[nuch] neurons and identified their direct postsynaptic targets. We found that the most strongly connected neurons, receiving up to 22 synapses from SN[nuch] cells, were two pairs of interneurons (IN[arc]) with a unique biramous morphology and contralaterally projecting axons (*Figure 10D–F*; *Figure 10—figure supplement 1*; *Video 4*). Two of the IN[arc] neurons were previously shown to be postsynaptic to the photoreceptor cells of the larval eyespots and have connections to the ventral motorneurons (*Randel et al., 2015*). This represents a potential functional path linking the nuchal organs to the locomotor apparatus. We also identified several other neurons that received a few synapses from the SN[nuch] cells, including two interneurons (IN[RGWa-dcr1], IN[RGWa-dcl1]) that were identified by siGOLD labeling to express the RGWa neuropeptide (*Figure 9A,B*). Additionally, 87 neurons received only one synapse from one of the SN[nuch] cells (*Figure 10—figure supplement 2*). The strong connectivity of the IN[arc] cells to the nuchal organ relative to these other neurons suggests that the IN[arc] cells represent the functionally relevant targets in the circuit.

Mapping the position of presynaptic sites of SN[nuch] cells revealed that these cells form synapses bilaterally (*Figure 10 H, I*). This is in contrast to the adult eye photoreceptor cells that only form ipsilateral synapses (*Figure 10 H, I*) (*Randel et al., 2014*).

To characterize the synapses of the PDF-positive SN[nuch] cells in more detail, we examined several synapses in high-resolution (2.22 nm/pixel) images (*Randel et al., 2015*). We found that SN[nuch] presynaptic sites contained only DCVs (*Figure 10J,K*). These synapses could be clearly distinguished from classical neurotransmitter synapses (*Figure 10L*) (*Randel et al., 2014*) based on the larger size of the synaptic vesicles (mean diameter=63 nm, S.D. = 8.4, n=100 vesicles) and the electrondense core of the vesicles (*Figure 10J,K*). Furthermore, mapping classical neurotransmitter markers by in situ hybridization did not reveal the presence of any classical neurotransmitters in the nuchal organ region (*Randel et al., 2014*). Although we did not directly immunogold label any of the SN[nuch] synapses, these observations suggest that the SN[nuch] neurons signal to their target interneurons by PDF-containing vesicles concentrated at peptidergic synapses.

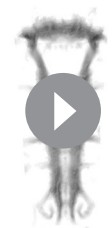

**Video 1.** Neuronal arrangements in the ventral nerve cord detected by whole-body IF and image registration. Ventral view of 72 hpf *Platynereis*. VNC (grey) of average whole-body reference template stained with acetylated tubulin, generated from scans of 36 individuals. Onto this reference scaffold, we project individual registered immunostaining patterns generated by the different neuropeptide antibodies. In order of appearance: FVa (orange), RGWa (yellow), LEUC (brown), FVRIa (lavender), RYa (forest green), LUQ (magenta), MIP (aqua), ATO (sky blue), FMRFa (lime green), PENK (red), PDF (royal blue).

## Discussion

In this paper, we introduced siGOLD, a method to molecularly identify specific neurons in large serial EM datasets based on the immunogold labeling of subsets of sections followed by serial reconstruction of the tagged neurons. Due to the sparse labeling of a few sections across many, siGOLD allows the use of several different antibody labels, each at different sections, within a single dataset. siGOLD relies on continuous series of sections of sufficient quality that allow the tracing of neurites in order to reconstruct the morphologies of labeled neurons.

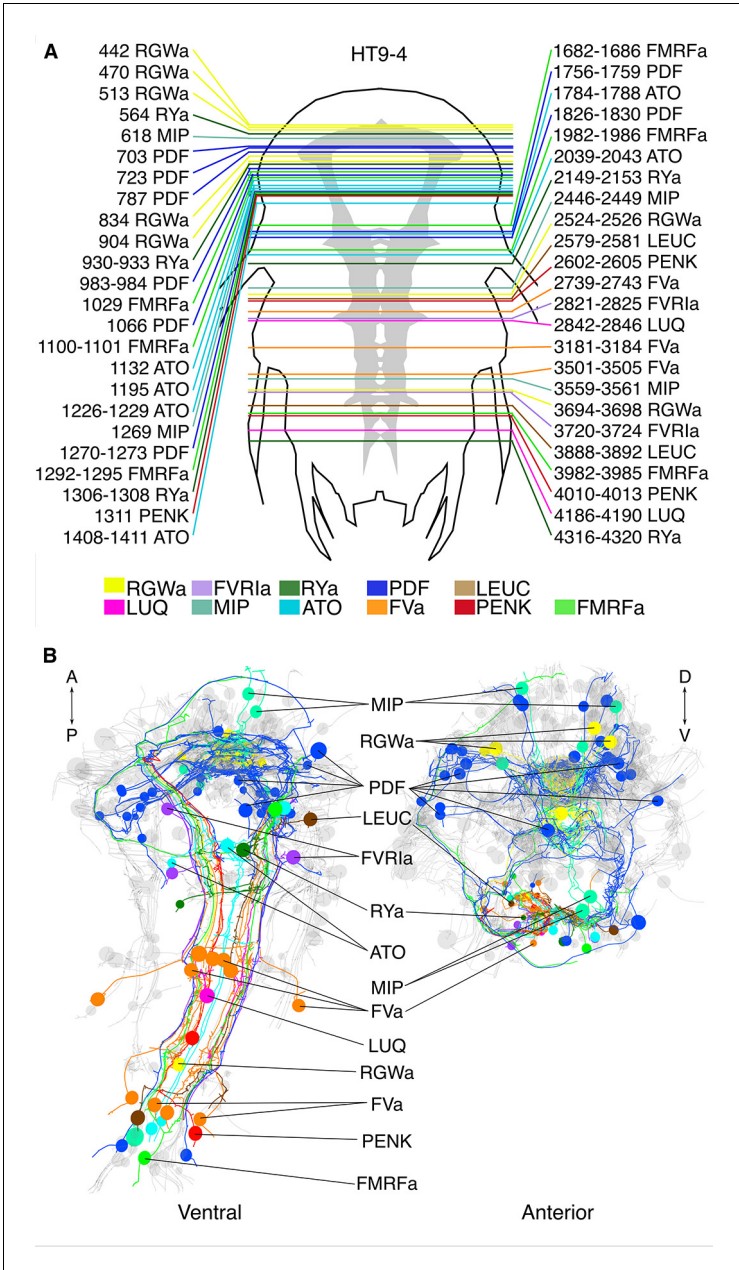

**Figure 6.** siGOLD labeling in a whole-body serial EM dataset (HT9-4). (**A**) Schematic of the HT9-4 specimen showing the position of the VNC (grey), ventral view. The entire larva was fully sectioned and imaged. Colored lines indicate the position of sections that were used for immunolabeling. For each line, layer number(s) followed by the name of the neuropeptide that was immunogold-labeled in those layers are indicated. (**B**) Ventral and anterior views of all fully traced immunogold-labeled peptidergic neurons. Detailed layer information is available in *Figure 6—source data 1*.

The following source data is available for figure 6:

**Source data 1.** Complete layer statistics of the sections and images of HT9-4.

Since siGOLD requires the sparse labeling of small subsets of sections, it is ideally used with markers that broadly label a neuron's morphology. We established siGOLD using neuropeptide antibodies for two reasons. First, neuropeptides are distributed along the entire length of axons, due to their active circulation throughout the axon (*Wong et al., 2012*). Second, neuropeptide antigens

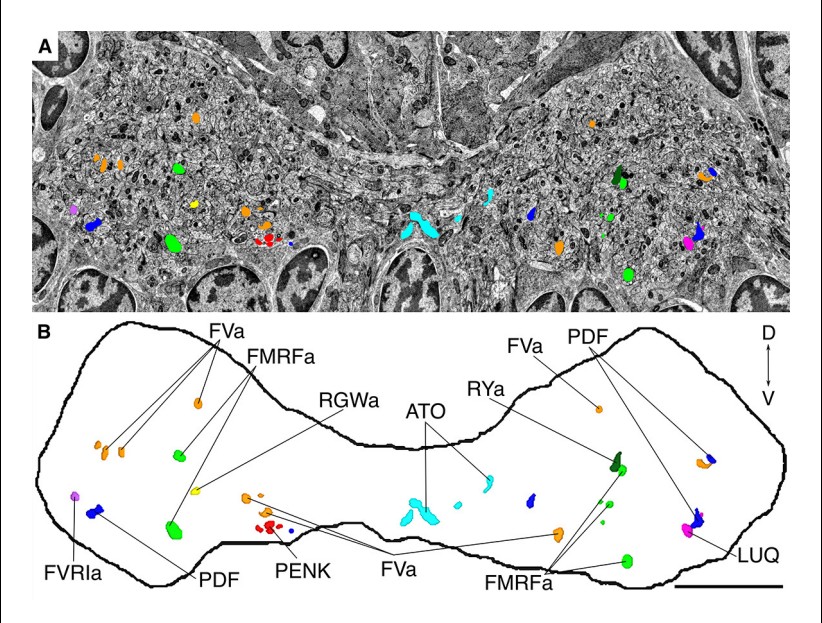

**Figure 7.** Position of siGOLD-labeled neurites in a cross-section of the VNC in HT9-4. (**A**) TEM image of a VNC cross-section with the segmented profiles of peptidergic neurites identified by siGOLD. Different colors represent different neuropeptide-antibody labeling. (**B**) Segmented profiles of peptidergic axons in a cross-section through the VNC near the base of the circumesophageal connectives at the first commissure. All traced neuropeptidergic axons crossing the VNC at that level in the HT9-4 specimen are shown. Scale bar: (**B**) 5 μm.

have unique properties, including their small size, high abundance, concentration in DCVs, and frequent C-terminal amidation, a modification that confers increased stability and immunogenicity to mature neuropeptides (*Conzelmann and Jékely, 2012*; *Eipper et al., 1992*). In agreement with the extreme stability of neuropeptide immunoreactivity, we could perform immunoEM of 'set-aside' layers using neuropeptide antibodies years after sectioning and the acquisition of the complete image series had been completed.

We did not test other antibodies, but several generally used antibody markers, including antibodies against neurotransmitters, transporters, or enzymes (*Collman et al., 2015*), could in principle be suitable for neuron identification using siGOLD. We established siGOLD on Epon-embedded samples that allowed robust sectioning of thousands of sections and provided high ultrastructural detail. Since Epon-embedding is known to compromise the immunogenicity of some antigens (*Brorson, 1998*; *Brorson and Reinholt, 2008*; *De Paul et al., 2012*), labeling for such antigens may require the use of alternative embedding resins such as Lowicryl HM-20 that provides excellent ultrastructural contrast and is compatible with many antibodies (*Collman et al., 2015*).

siGOLD is compatible with conventional TEM using sections collected on slotted, plastic-coated nickel grids (*Skepper and Powell, 2008*). It is also compatible with sectioning methods that collect long ribbons of sections on a glass slide (*Blumer et al., 2002*) or the high-throughput automatic tape-collecting ultramicrotome (ATUM) method (*Hayworth et al., 2014*). However, siGOLD is not compatible with sectioning procedures that destroy the sections, such as focused ion beam (FIB) SEM (*Reyntjens and Puers, 2001*) or the serial block-face method (*Denk and Horstmann, 2004*).

The siGOLD method differs from conjugate light-electron array tomography (*Collman et al., 2015*) and CLEM in several aspects. Array tomography was developed to provide detailed molecular profiling of individual synapses, and uses repeated immunolabeling of every section. siGOLD does not rely on the staining of every section with multiple markers, but rather the staining of sparsely distributed sections with one marker each. The aim of siGOLD is to assign molecular tags to several different neurons and to trace them through many layers. Array tomography and CLEM involves the registration of separately acquired IF and EM images, and the relocation of the sample between the different imaging setups can be challenging and time consuming (*Timmermans and Otto, 2015*). In

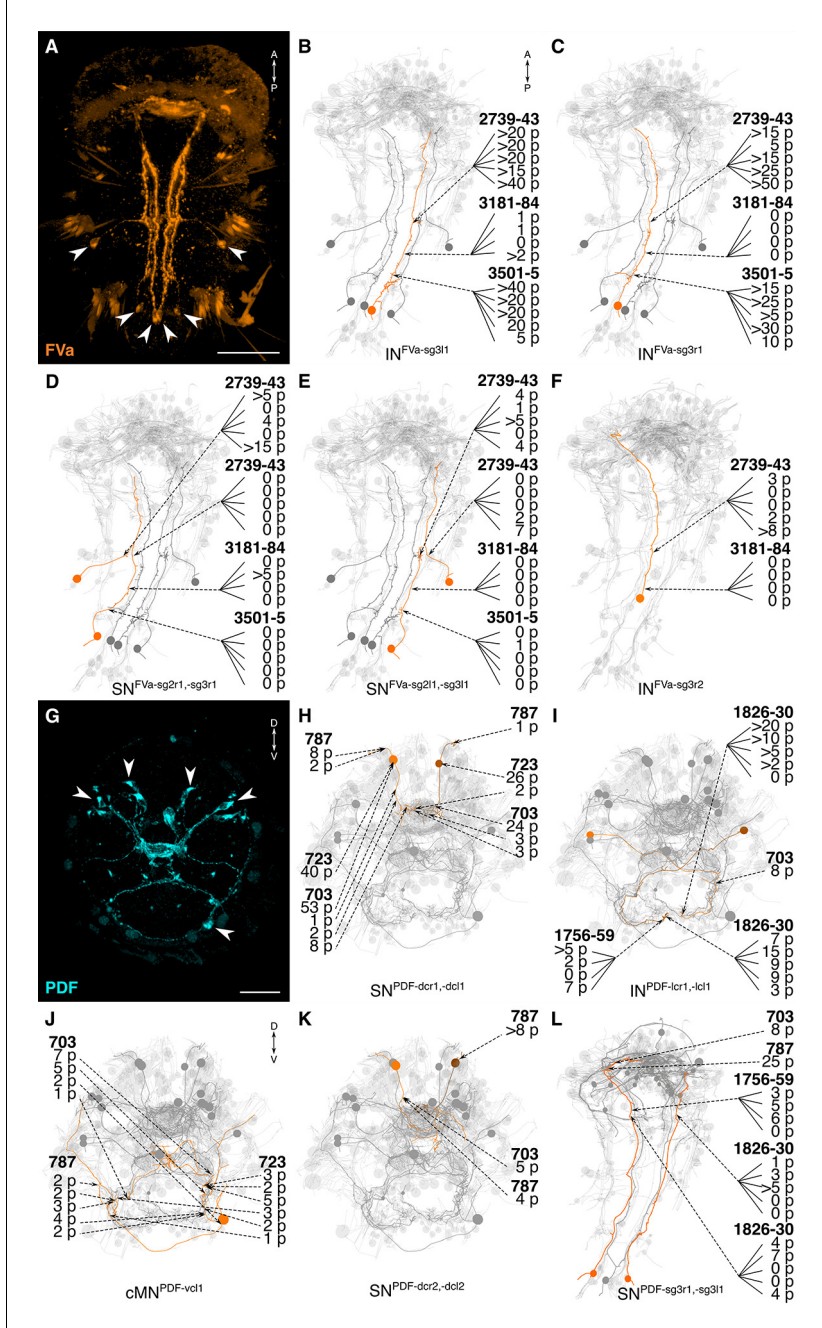

**Figure 8.** siGOLD labeling and whole-body neuron reconstructions in HT9-4. (**A, G**) Full body IF labeling of FVa and PDF-positive cells, ventral view and anterior view, respectively. Note that (**A**) is 72 hpf and (**G**) is 48 hpf. Arrowheads point to neuron cell-bodies that were traced in EM reconstructions. (**B–F, H–L**) Fully reconstructed neurons, identified using the siGOLD method in the full body HT9-4 dataset. Dashed arrows indicate immunogold labeled layers along the neurite, layer number(s) and number of gold particles per layer are shown. Reconstructed FVa (**B–F**) and PDF (**H–L**) positive cells. Scale bars: (**A**) 50 μm; (**G**) 30 μm.

contrast, siGOLD relies on the direct immunoEM labeling of sections and no registration step is needed. With siGOLD, we enjoy the full resolving power of the electron microscope, and in this study we could also identify individual DCVs that carry strong immunogold signal in the neurite profile of specific peptidergic neurons.

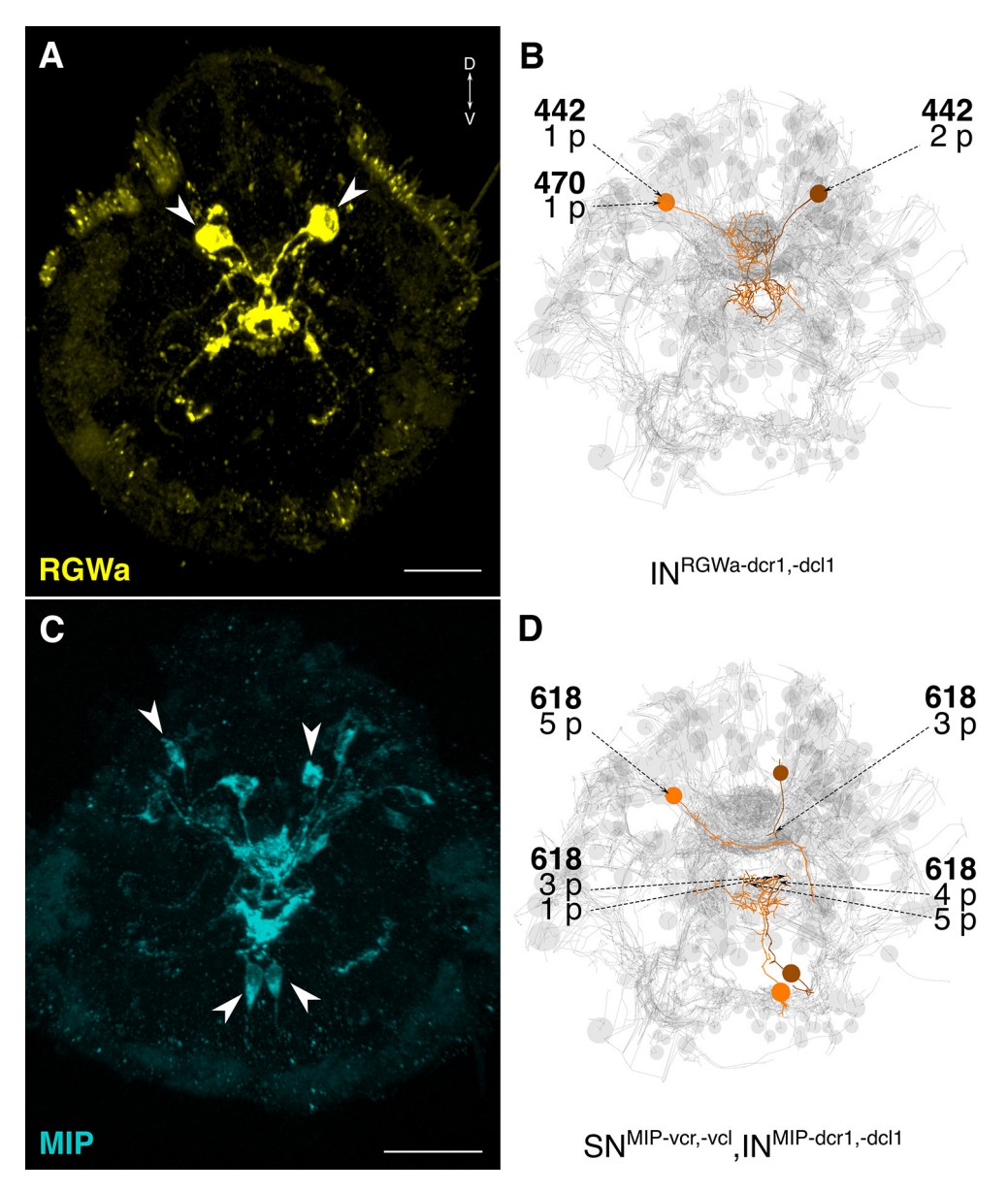

**Figure 9.** siGOLD labeling and whole-body neuron reconstructions in HT9-4. (**A, C**) Full body IF labeling of RGWa and MIP-positive cells in 72 hpf larvae, anterior views. Arrowheads point to neurons that were traced in EM reconstructions. (**B, D**) Traced neurons, identified using the siGOLD method in the full body HT9-4 dataset. Dashed arrows indicate immunogold-labeled layers along the neurite, layer number(s) and the number of gold particles per layer are shown. Reconstructed RGWa (**B**) and MIP (**D**) positive cells. Scale bars: (**A, C**) 30 μm.

One shortcoming of the immunoEM approach is that sections cannot be relabeled after they have been contrasted and exposed to the electron beam. However, given that in siGOLD only a subset of the sections is processed for immunoEM, it is possible to use an arbitrarily large number of antibodies at different sections from a series that encompasses a large volume of tissue, allowing the multiplex identification of neuron types.

We demonstrated that siGOLD can be used to molecularly identify neurons in large serial EM datasets, in conjunction with the reconstruction of the synaptic connectivity of these neurons. We reconstructed the sensory neurons of the nuchal organs and their postsynaptic partners, representing a candidate chemotactic circuit in the *Platynereis* larval head. siGOLD labeling revealed that the

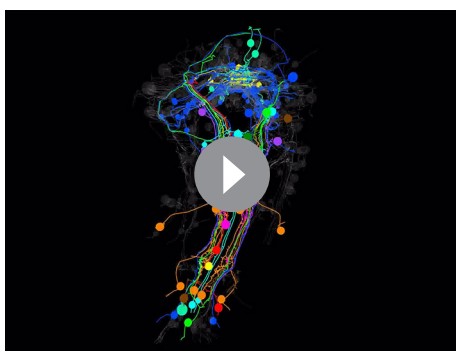

**Video 2.** 3D view of the 72 hpf HT9-4 larva showing fully-traced peptidergic neurons identified by siGOLD. Ventral view of the 72 hpf *Platynereis* siGOLD-tagged peptidergic neurons are shown in the following colors: FVa (orange), RGWa (yellow), LEUC (brown), FVRIa (lavender), RYa (forest green), LUQ (magenta), MIP (aqua), ATO (sky blue), FMRFa (lime green), PENK (red), PDF (royal blue). Other non-labeled traced neurons (grey) provide the outline of the larva.

sensory neurons express the PDF neuropeptide. The presence of DCVs and lack of clusters of clear vesicles at presynaptic sites in these neurons suggest the presence of peptidergic transmission to the target interneurons.

The bilateral projection of the nuchal organ sensory neurons is similar to that seen for olfactory receptor neurons in *Drosophila*. These neurons form bilateral projections, and the lateralization of odor processing is achieved by an asymmetry in neurotransmitter release (*Gaudry et al., 2013*). The nuchal organ circuit may use a similar computational strategy, and this is likely different from the processing of visual information (*Randel et al., 2014*; *Randel et al., 2015*). The molecular information obtained by siGOLD provides a useful entry point to the genetic investigation of this putative chemotactic circuit in *Platynereis*.

The mapping of peptidergic neurons by siGOLD, together with the recent identification of several neuropeptide receptors in *Platynereis* (*Bauknecht and Jékely, 2015*), open up the possibility of a cellular-level analysis of peptidergic neurotransmission in diverse circuits in *Platynereis*. In total, we have identified 83 different peptidergic neurons out of approximately 2000 total in the *Platynereis* whole-body dataset. Further connectome tracing in these data will provide a rich source of information about specific peptidergic circuit motifs in this animal.

The siGOLD approach and the use of neuropeptide antibodies could also be adapted to enrich connectome data with molecular information in other organisms. In *Platynereis*, as well as vertebrates, *C. elegans*, and *Drosophila*, the majority of mature neuropeptides are amidated (*Jékely, 2013*; *Mirabeau and Joly, 2013*), and several antibodies are available (e.g. (*Johard et al., 2008*; *Nässel, 1993*)) or could readily be generated. Some neuropeptide antibodies have already been shown to work with immunoEM. For example, PDF neuropeptide-containing neurons were identified in the *Drosophila* brain by immunoEM and their local synaptic inputs were reconstructed from a series of sections (*Yasuyama and Meinertzhagen, 2010*). This and other antibodies could be used for siGOLD labeling in larger-scale connectome projects in *Drosophila*. Similar protocols and reagents could also be established for other organisms. siGOLD could also be adapted to organisms where transgenic tools to deliver EM-compatible markers are not available. In such organisms, the use of cross-species antibodies (*Conzelmann and Jékely, 2012*; *Nässel, 1993*) is a promising approach that would also allow the comparison of specific peptidergic neurons and their circuits across species.

The direct overlaying of chemical neuromodulatory maps onto synaptic connectomic maps by siGOLD opens up new possibilities for the study of nervous systems.

## Materials and methods

### Transmission electron microscopy

Fixation and embedding was carried out on 72 hpf *Platynereis* larvae (HT9-4 and HT9-5) as described previously (*Conzelmann et al., 2013a*). Forty nanometer serial sections were cut on a Reichert Jung Ultracut E microtome using a 45° DiATOME Diamond knife. The sections were collected on single-slotted copper grids (NOTCH-NUM 2_1 mm, Science Service, Munich) with Formvar support film. For the HT9-5 serial sections, the samples were collected on single-slotted nickel grids coated with Formvar support film. The ribbons of sections were picked up from underneath and on the underside (the notch side) of the grids. This method allowed the section ribbons to stay afloat and stretch while being dried in the oven, eliminating most wrinkles from the sections. The section statistics for

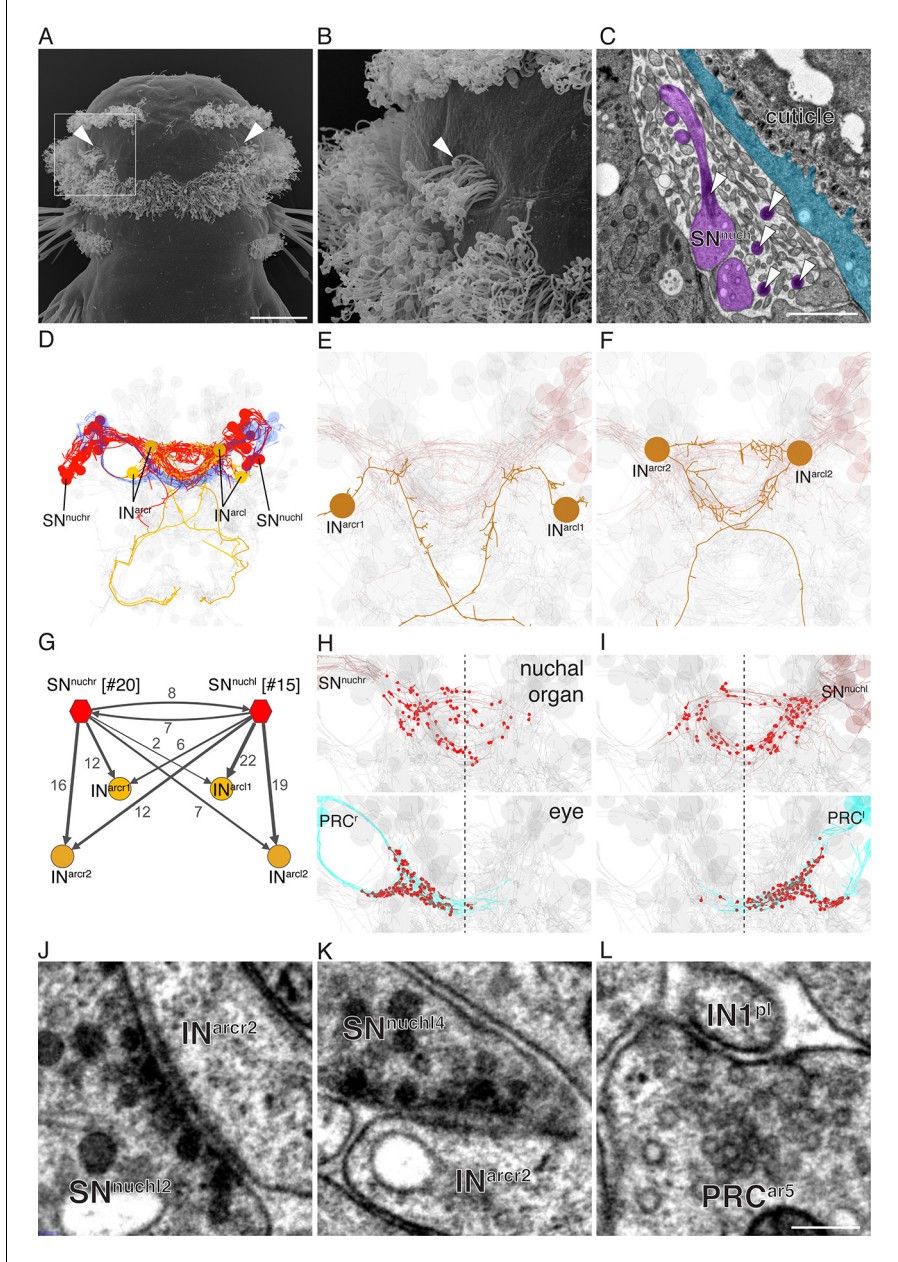

**Figure 10.** Reconstruction of a siGOLD-labeled peptidergic circuit. (**A, B**) SEM image of the nuchal organs (arrowhead) in the dorsal-posterior head of a 72 hpf *Platynereis* larva, dorsal view. Boxed area is shown enlarged in (**B**). Arrowheads point at motile cilia above the olfactory pit. (**C**) TEM image of a cross section of the nuchal organ showing the olfactory pit with sensory neurons and microvilli. Arrowheads point at sensory cilia. Sensory endings in the olfactory pit are highlighted in magenta, an epithelial cell underneath the cuticle is highlighted in blue. (**D**) 3D reconstruction of the nuchal organ circuit. SN$^{nuch}$ neurons (red), and IN$^{arc}$ (orange) interneurons are shown. The photoreceptor cells of the adult eyes are shown in blue as a reference. (**E, F**) 3D reconstruction of IN$^{arc}$ (orange) interneurons. (**G**) Graph representation of SN$^{nuch}$ connectivity. Nodes represent neurons or groups of neurons, edges represent synaptic connections. The number of synapses is indicated on each arrow. Edge thickness is proportional to the square root of the number of synapses. (**H, I**) Presynaptic sites (red dots) in SN$^{nuch}$ neurons and adult-eye photoreceptor cells (PRC) with their soma on the right (**H**) or left (**I**) side of the body. (**J, K**) Peptidergic synapses in SN$^{nuch}$ neurons. (**L**) A glutamatergic synapse in an adult eye photoreceptor cell. Scale bar: (**A**) 30 μm; (**C**) 1 μm; (**L**) 150 nm.

The following figure supplements are available for figure 10:

*Figure 10 continued on next page*

*Figure 10 continued*

**Figure supplement 1.** Morphology of SN$^{nuch}$ and IN$^{arc}$ neurons.
**Figure supplement 2.** Number of synaptic inputs from SN$^{nuch}$ cells to all postsynaptic targets.

the HT9-4 specimen (or NAOMI) were previously described (*Randel et al., 2015*) and an updated table with the immunoEM information is shown in *Figure 6—source data 1*. For HT9-5, 200 sections were cut from the VNC (first segment) and imaged for tracing, segmentation and analysis.

## Grid handling

To meet the demand for consistent section processing in large-scale serial reconstruction projects, we have developed an effective method of grid handling based on a method first described in *Rowley and Moran (1975)*. The following method has proven safe for contrast staining, immunolabeling, and carbon coating. It stabilizes the grids during the staining procedures and saves time by eliminating the need for one-by-one grid manipulation.

A 6 × 12 hole microwell mini tray plate (NUNC™ Brand MicroWell® Mini Trays) was cut into two 3 × 12 hole plate strips. Holes were drilled through each of the plate's micro-cups using a fine drill to later allow the plastic support film to dry evenly on both sides. Plates were rinsed with 75% ethanol, followed by distilled water, then sonicated to eliminate dust and burr particles. This procedure produced two lightweight plates similar to that described in *Rowley and Moran (1975)* with the additional benefit of 'raised', separated walls. This approach prevents cross-contamination by diffusion of solutions during the staining and labeling procedures. A strip of Formvar support film was used to coat the microwell mini plate on the topside of the plate. A small water droplet (10 µl) was placed on each hole of the plate on the support film using a filtered syringe. A grid containing previously cut sections was placed (sections facing up) on each droplet and dried in the 50°C oven until the grids and the plastic support film had fused. Grids were then ready for immunolabeling and contrast staining by inverting and matching to the micro-cups of a full mini tray with desired solutions (*Figure 1A* – Mini tray).

**Table 2.** Number of gold particles in SN$^{nuch}$ neurons in three layers labeled with the PDF antibody.

|  | Layer | | | | Layer | | |
|---|---|---|---|---|---|---|---|
| SN$^{nuch}$ | 703 | 723 | 787 | SN$^{nuch}$ | 703 | 723 | 787 |
| r1 | 0 | 2 | 0 | l1 | 0 | 0 | 0 |
| r2 | 0 | 0 | 0 | l2 | 1 | 0 | 4 |
| r3 | 0 | 0 | 0 | l3 | 0 | 0 | 0 |
| r4 | 0 | 0 | 7 | l4 | 0 | 0 | 0 |
| r5 | 0 | 0 | 0 | l5 | 0 | 0 | 0 |
| r6 | 2 | 0 | 0 | l6 | 0 | 3 | 10 |
| r7 | 0 | 0 | 0 | l7 | 0 | 0 | 2 |
| r8 | 0 | 1 | 2 | l8 | 1 | 0 | 3 |
| r9 | 0 | 0 | 0 | l9 | 0 | 0 | 0 |
| r10 | 0 | 0 | 0 | l10 | 0 | 0 | 0 |
| r11 | 0 | 0 | 0 | l11 | 0 | 0 | 0 |
| r12 | 0 | 3 | 0 | l12 | 6 | 1 | 0 |
| r13 | 1 | 3 | 2 | l13 | 9 | 9 | 6 |
| r14 | 0 | 0 | 0 | l14 | 18 | 0 | 0 |
| r15-r20 | 0 | 0 | 0 | l15 | 0 | 0 | 0 |

Number of gold particles in SN$^{nuch}$ neurons identified in three different immunogold layers labeled with the PDF antibody in all SN$^{nuch}$ sensory neurons.

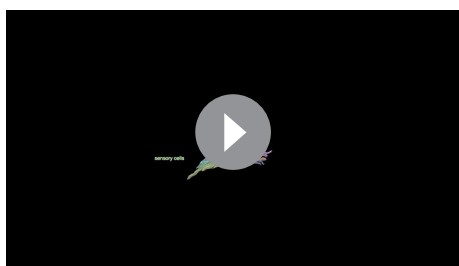

**Video 3.** Reconstruction of the nuchal organ. Reconstruction of the nuchal organ from the dorsal head of the 72 hpf *Platynereis* larva dataset HT9-4. The nuchal organ consists of three multiciliated cells and 16 sensory cells which project a sensory cilium and branching microvilli into an olfactory pit just below the cuticular layer.

## Contrasting

Dried sections were contrast stained with drops of 2% aqueous uranyl acetate for 6 min. Grids were washed by gently lowering the entire plate into a clean glass beaker of filtered distilled water. Excess water was wicked away with filter paper, and the grids were dried at 50°C for 1–2 min. Next, the grids were stained with drops of Sato's triple lead stain (*Hanaichi et al., 1986*) for 4 min. Grids were washed and dried as described above and dried at 50°C for 1 min. Grids were then carbon coated before being imaged with a FEI TECNAI Spirit electron microscope.

## Immunogold silver-enhancement method

For immunoEM, we have eliminated the etching step prior to primary antibody incubation. We have observed that the treatment with the etching solution destroys ultrastructural integrity. In our technique, the use of sheep serum and bovine serum albumin mixed with Tween-20 at pH of 7.9 greatly enhanced the immunogold signal.

The following protocol was performed over a 1-day period and is an optimized version of a protocol by Stierhof (*Stierhof et al., 1991*) for Epon-embedded sections. All incubations up to the silver enhancement point were done in the TBST-BGN (Tris buffer saline with Tween-20, BSA, Fish Gelatin and Normal Sheep Serum, pH 7.9; see section below on preparing the buffer). First, sections were blocked with TBST-BGN for 10 min; blocked sections were then incubated with primary antibody (1:25 dilution) in TBST-BGN for 2 hr. Grids were washed twice with TBST-BGN for 5 min, and then incubated with secondary goat anti-rabbit IgG antibody with gold conjugates (AURION Ultra Small Immunogold Reagents; size 0.8 nm) at 1:50 dilution for 1 hr. Grids were then washed with TBST-BGN twice for 5 min, then washed twice with filtered distilled water for 5 min, followed by fixation with 1% glutaraldehyde in filtered distilled water for 5 min. Sections were washed in filtered distilled water twice for 5 min. Silver enhancement was applied with Aurion silver-enhancement kit for up to 47 min. The time interval of silver enhancement was dependent on temperature and concentration of the silver-enhancement solution. Gold size growth was approximately 15–20 nm within this time interval, a size appropriate for observation of gold particles in EM. Silver enhancement was stopped with three 3-min washes in filtered distilled water. Excess water was gently wicked from the back of the plate and individual holes with wedges of Whatman filter paper. Grids were dried in a 50°C oven and then contrasted (see above).

## Preparation of blocking and wash buffer solution

Buffer concentration and consistency is very important in immunological reactions. We compared many protocols to achieve the best buffer for our experiments. We found that a saline buffer with Tris, Tween, bovine serum albumin (BSA), fish Gelatin, and normal serum (TBST-BGN), worked well. Our solutions were always freshly prepared on the day of the experiment.

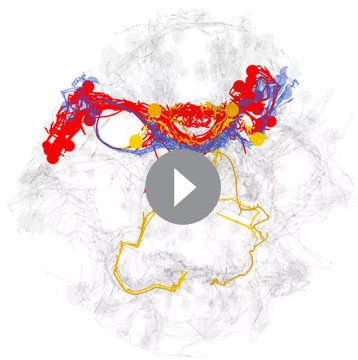

**Video 4.** Reconstruction of the nuchal organ circuit. The nuchal organ sensory cells (SN[nuch], red) connect to two pairs of IN[arc] interneurons (orange). The adult eye photoreceptors (dark blue) are shown for reference. Several other neurons are shown in pale grey to highlight the shape of the larval nervous system.

To prepare the buffer for immunoEM, three separate solutions were made: (1) 0.61 g Trizma, 0.90 g NaCl, and 70 ml of distilled water, (2) 1.0 g of BSA, 1.5 ml normal sheep serum, 500 µl Tween-20, and 20 ml distilled water, and (3) 0.01 g of fish Gelatin and 10 ml of distilled water. Solution 3 was heated to dissolve and then cooled before being added to solutions 1 and 2. Finally, solutions 1–3 were combined and gently mixed. The buffer solution was then adjusted to pH of 7.9 using 1 M HCl and syringe filtered with a 0.45 µm filter.

## Imaging and post-processing

Image acquisition of TEM serial sections was performed on a FEI TECNAI Spirit transmission electron microscope equipped with an UltraScan 4000 4X4k digital camera using the image acquisition software Digital Micrograph (Gatan Software Team Inc., Pleasanton) and SerialEM (*Mastronarde, 2005*). The images for the HT9-4 and HT9-5 samples were scanned at a pixel resolution of 5.71 nm/pixel and 2.22 nm/pixel, respectively. Image stitching and alignment were accomplished using TrakEM2 (*Cardona et al., 2010*; *Cardona et al., 2012*). All structures were segmented manually as area-lists, and exported into 3Dviewer and Blender as previously described (*Asadulina et al., 2015*). Tracing and annotation of the connectome were performed with CATMAID, a collaborative annotation toolkit for large-scale image data (*Saalfeld et al., 2009*; *Schneider-Mizell et al., 2015*).

## IF and image registration

Antibodies were generated by immunizing rats or rabbits with synthetic, amidated peptides. Some of the antibodies were used in previous studies (*Conzelmann and Jékely, 2012*; *Conzelmann et al., 2011*; *Conzelmann et al., 2013a*; *Jékely et al., 2008*). All peptides contained an N-term Cys that was used for coupling. Antibodies were affinity purified from sera as previously described (*Conzelmann and Jékely, 2012*). Immunostainings were carried out as previously described (*Conzelmann and Jékely, 2012*). For triple IF, we used secondary antibodies coupled to three different fluorophores (FVa rat, AF488 secondary antibody rat; other neuropeptide rabbit, AF647 secondary antibody rabbit; acetylated-tubulin mouse, high-fidelity AF555 secondary antibody mouse).

For image registration, an average full-body acetylated tubulin reference template was generated for 72 hpf *Platynereis* using a modification of a previously described method (*Figure 5—source data 1*) (*Asadulina et al., 2012*). We used full-body scans of 36 larvae generated on a Zeiss LSM 780 confocal microscope with ZenBlue software. All stacks were oriented and their centers of mass were aligned. Stacks were averaged, and all stacks were aligned to this first average template using affine transformation. The 24 registered stacks most similar to the first average (as determined by an iteration metric) were then used to create an affine-transformed average. All original oriented stacks were then aligned to the affine-transformed average using affine and deformable transformation. The 24 registered stacks most similar to the affine-transformed average were then used to create an affine/deformable-transformed average. All original oriented stacks were then aligned to the affine/deformable-transformed average using affine and deformable transformation. The 24 registered stacks most similar to the affine/deformable-transformed average were then used to generate the final average acetylated tubulin reference template. The final average stack was unbiased and was used for image registration.

Imaris was used for image processing (adjusting brightness and contrast uniformly) and to take virtual cross-sections of the VNC for comparison with siGOLD samples.

## Morpholino injection

Morpholino injections were performed as previously described (*Conzelmann et al., 2013a*). We used the following morpholinos to target the various neuropeptide precursor genes (GeneTools, LLC): Pdu-FMRFa-start MO CCACTGGTCCCTCATGGCAGGGTTT, Pdu-PDF-start MO CTGAACTGTCTTGCTT-GATCCCATC, Pdu-ATO-start MO CACAGGACTACCTTCATTTTTCTGA, Pdu-LUQ-start MO GTATT-TGCACAACATAGTGATAGTC, Pdu-PENK-start MO GAGGAGGACCACCAATATCTTCATC, Pdu-RYa-start MO TATAGACATGACACCTTGTTGGAGT, Pdu-LEUC-start2 MO TCTTGGCTGAACTCATT-GCGGCC, Pdu-FVa-start MO CCATCCGCCCACGCTCATATGCATC, Pdu-FVRIa-start MO CCCCCTT-CATACTGTCACAACGGAC, Pdu-RGWa-start MO CGACGACCCCCTGTAGCTTCATGTC.

## Acknowledgements

We thank Matthias Flötenmeyer and Heinz Schwarz for advice on immunogold labeling and EM, Nadine Randel for help with tracing, Stephan Saalfeld for advice on the generation of an unbiased template for image registration, David Mastronarde for developing serialEM, Steffen Schmidt for maintaining our Catmaid server, Aurora Panzera for help with microinjections, and Dorothee Hildebrandt for animal care. The research leading to these results received funding from the European Research Council under the European Union's Seventh Framework Programme (FP7/2007-2013)/ European Research Council Grant Agreement 260821. This project is supported by the Marie Curie ITN "Neptune", GA 317172, funded under the FP7, PEOPLE Work Programme of the European Commission.

## Additional information

### Funding

| Funder | Grant reference number | Author |
|---|---|---|
| European Research Council | 260821 | Réza Shahidi<br>Elizabeth A Williams<br>Markus Conzelmann<br>Albina Asadulina<br>Csaba Verasztó |
| Max-Planck-Gesellschaft | | Luis A Bezares-Calderón<br>Gáspár Jékely |
| European Commission | GA 317172 | Sanja Jasek |

The funders had no role in study design, data collection and interpretation, or the decision to submit the work for publication.

### Author contributions

RS, Immunogold microscopy, Conception and design, Acquisition of data, Analysis and interpretation of data, Drafting or revising the article; EAW, Morpholino injections, IF, image registration, Conception and design, Acquisition of data, Analysis and interpretation of data, Drafting or revising the article; MC, Antibody generation, Acquisition of data, Analysis and interpretation of data; AA, Development of image registration protocol, Conception and design, Analysis and interpretation of data; CV, SJ, LABC, EM tracing, Analysis and interpretation of data; GJ, Conception and design, Analysis and interpretation of data, Drafting or revising the article

### Author ORCIDs

Gáspár Jékely, http://orcid.org/0000-0001-8496-9836

## Additional files

### Major datasets

The following datasets were generated:

| Author(s) | Year | Dataset title | Dataset URL | Database, license, and accessibility information |
|---|---|---|---|---|
| Shahidi R, Williams EA, Conzelmann M, Asadulina A, Verasztó C, Jasek S, Bezares-Calderón LA, Jékely G | 2015 | Data from: A serial multiplex immunogold labeling method for identifying peptidergic neurons in connectomes | http://dx.doi.org/10.5061/dryad.c7366 | Available at Dryad Digital Repository under a CC0 Public Domain Dedication. |

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
