## [Decision Letter]

Thank you for submitting your work entitled "siGOLD: a Serial Multiplex Immunogold Labeling Method for Identifying Peptidergic Neurons in Connectomes" for consideration by *eLife*. Your article has been reviewed by Eve Marder (Senior Editor) and three peer reviewers, one of whom, Ronald L. Calabrese, is a member of our Board of Reviewing Editors.

The reviewers have discussed the reviews with one another and the Reviewing editor has drafted this decision to help you prepare a revised submission.

Summary:

The authors report a new technique that allows overlaying peptidergic neuromodulatory maps onto EM level synaptic connectivity maps in the study of nervous systems. This technique uses serial multiplex immunogold labeling (siGOLD) combined with standard serial-section transmission electron microscopy. Immunogold labeling for amidated neuropeptide epitopes, which show robust and long-term immunopreservation in Epon-embedded samples, is used, one marker per section, on sparsely distributed sections. Peptides provide unique markers because they are widely distributed in neurons including along the full lengths of axons. The advantage of siGOLD is that it relies on direct immunoEM of sections and does not rely on registering EM and IF images to assign immunolabels to specific structures. They use this technique to identify peptidergic neurons for eleven different neuropeptides in the CNS of 72 hpf *Platynereis* larva and they also reconstruct part of a sensory peptidergic circuit in the head of this larva.

The strength of the manuscript is the novelty and clearly demonstrated utility of the technique and its potential applicability to other preparations that are actively under investigation, including but not limited to mouse and zebra fish. Importantly, it can be used for organisms that are not now amenable to facile genetic manipulations.

Essential revisions:

The major weaknesses pointed out in the expert reviews and which require revision are:

1) A poor presentation in which the excitement of the work is not conveyed and is irregular at places. Additionally, the choice of the sensory nuchal organs for illustration of the method is neither introduced nor discussed. There is presumably considerable scientific interest in these findings but these are not highlighted.

2) The authors do not do a thorough enough comparison with related methods (e.g. superresolution microscopy to image fluorescent probes and correlating their location with EM and array tomography) nor do they discuss applicability to other organisms adequately.

Below are excerpts from the expert reviews that expand on these needed revisions and point out other more minor revisions and clarifications that are needed.

A) It isn't clear how generalizable the method is to other targets beyond neuropeptides. Do other common neuro-related targets, such as receptors, enzymes, calcium binding proteins, etc. also survive Epon embedding so that etching steps are not required? It doesn't necessarily detract from the study if only neuropeptides can be labeled using the protocol described, but it would be helpful to discuss how broadly applicable is the approach.

B) We had a question about the statement 'Neurites that were labeled on one grid were also strongly labeled on their grid pair with the same antibody separated by approximately 50 sections.' Examination of Figure 2—source data 1 seems to indicate violations of this statement. For example, n13 labels for LUQ in layers 44-50, but not in layers 1-5. Also, n5 labels for cFVa in layers 18-23, but not in 69-74. And so on. Are we missing something? Were the axons just not contained in the unlabeled layer ranges? Or did the axons not contain dense core vesicles in the unlabeled ranges?

C) Was the average distance between dense core vesicles along axons taken in to consideration when deciding how to space out the immuno-labeled sections in the whole body EM dataset? Is there anything that prevents immuno-labeling every section besides time and effort?

D) The circuit diagram in Figure 10 seems to imply the synapses indicated by the directed graph edges are using the respective neuropeptides for transmission. But this point isn't explicitly made. Were the immuno-labeled dense core vesicles always associated with a clear postsynaptic partner?

E) The introduction on the topic of Epon resin transitions abruptly. A sentence stating that the limitations of Epon don't apply to neuropeptides and therefore this could be exploited is called for.

F) The Introduction does not touch at all on the nuchal system and the PDF-expressing neurons. What was the point of studying them? Why were they chosen as a case study?

G) In Figure 2, contrast seems optimized to detect immunogold particles. Processing with CLAHE would better reveal cytoplasmic membranes while still preserving the saliency of immunogold particles. What was the algorithm chosen for improving contrast?

H) In Figure 10, are the 1-synapse edges meaningful? What is the error rate and synapse count variance when redundantly or repeatedly mapping connections between neurons?

I) The manuscript does not address the advantage over array tomography: far faster imaging and higher resolution, both in Z (thinner sections) and in XY. "With siGOLD we enjoy the full resolving power of the electron microscope" should mention transmission EM; array tomography uses SEM which offers far slower image acquisition.

J) The Discussion does not address the issue of what has been learned by performing these labelings in *Platynereis* beyond merely proposing a circuit for chemotaxis.

---

## [Author Response]

*The major weaknesses pointed out in the expert reviews and which require revision are: 1) A poor presentation in which the excitement of the work is not conveyed and is irregular at places. Additionally, the choice of the sensory nuchal organs for illustration of the method is neither introduced nor discussed. There is presumably considerable scientific interest in these findings but these are not highlighted.*

We have thoroughly rewritten the text. We now introduce the nuchal organs in more detail, and discuss the interest of the findings. We have also expanded the figure describing the nuchal organ circuitry, and added more detail about the nature and distribution of synapses in the nuchal organ.

*2) The authors do not do a thorough enough comparison with related methods (e.g. superresolution microscopy to image fluorescent probes and correlating their location with EM and array tomography) nor do they discuss applicability to other organisms adequately.*

We have extended the Discussion and Introduction and compare array tomography, CLEM and siGOLD in more detail. We also discuss the cross-species applicability of the siGOLD technique in the revised version.

*Below are excerpts from the expert reviews that expand on these needed revisions and point out other more minor revisions and clarifications that are needed. A) It isn't clear how generalizable the method is to other targets beyond neuropeptides. Do other common neuro-related targets, such as receptors, enzymes, calcium binding proteins, etc. also survive Epon embedding so that etching steps are not required? It doesn't necessarily detract from the study if only neuropeptides can be labeled using the protocol described, but it would be helpful to discuss how broadly applicable is the approach.*

We did not test other targets beyond neuropeptides. We find it unlikely that receptors or enzymes would survive the Epon embedding procedure. We discuss the use of alternative embedding resins, such as Lowicryl HM-20 that provide excellent ultrastructural contrast and is compatible with many antibodies.

*B) We had a question about the statement 'Neurites that were labeled on one grid were also strongly labeled on their grid pair with the same antibody separated by approximately 50 sections.' Examination of Figure 2—source data 1 seems to indicate violations of this statement. For example, n13 labels for LUQ in layers 44-50, but not in layers 1-5. Also, n5 labels for cFVa in layers 18-23, but not in 69-74. And so on. Are we missing something? Were the axons just not contained in the unlabeled layer ranges? Or did the axons not contain dense core vesicles in the unlabeled ranges?*

In order to address this comment we have now performed further quantifications. We scored the number of dense core vesicles per slice in several axons in our test series (HT9-5). We found that the lack of consistent gold labeling of the same axon with the same antibody on grid pairs was always due either to the lack of vesicles in the non-labeled part of the axon, or the termination of the axon before the labeled sections. In these axons, we never found gold labeling in layers where the respective axons lacked dense-core vesicles. For example, n5 in layers 18-23 contains many dense-core vesicles, and labels strongly for cFVa, but it does not contain any vesicles in layers 69-74, and consistently lacks cFVa labeling. We could also demonstrate the non-uniform distribution of dense-core vesicles along peptidergic axons at the sub-micron scale. Despite this, a large number of sections contain dense-core vesicles meaning that the random selection of sections for gold labeling is a feasible approach with a high rate of success. We have now included these new data in Figure 3—figure supplement 1 and [Supplementary-material SD1-data].

*C) Was the average distance between dense core vesicles along axons taken in to consideration when deciding how to space out the immuno-labeled sections in the whole body EM dataset? Is there anything that prevents immuno-labeling every section besides time and effort?*

No, we picked sections at uniform intervals across the whole series. Regarding dense core vesicle distribution, see our response to comment B.

There is in principle nothing that prevents immunolabeling every section. However, besides time and effort the immunolabeling procedure introduces extra risks (losing or damaging sections), and reduces the contrast, making tracing more difficult. More importantly, immuno-labeling of every section is not necessary if the only goal is to obtain the peptidergic identity of many neurons in a series (our main objective with siGOLD). We now discuss this issue in the paper.

*D) The circuit diagram in Figure 10 seems to imply the synapses indicated by the directed graph edges are using the respective neuropeptides for transmission. But this point isn't explicitly made. Were the immuno-labeled dense core vesicles always associated with a clear postsynaptic partner?*

We have now characterized the synapses in the PDF positive nuchal organ sensory neurons in more detail. We measure the diameter of several vesicles that were associated with a postsynaptic partner. The vesicles are much larger than previously characterized glutamatergic synaptic vesicles of the eye photoreceptor cells. We also found that the presynaptic vesicles in the PDF neurons were dense core vesicles. These observations suggest that these neurons use neuropeptides for synaptic transmission. We have incorporated these data, including high-resolution images of example synapses (Figure 10).

Unfortunately, we did not directly gold label any synapse. Given that we only stained three sections with the PDF antibody in the head, the chances of finding a synapse exactly in these layers was slim. We labeled dense core vesicles that were not adjacent to the membrane, presumably representing the circulating pool of vesicles in the axon.

E) The introduction on the topic of Epon resin transitions abruptly. A sentence stating that the limitations of Epon don't apply to neuropeptides and therefore this could be exploited is called for.

We have now changed the Introduction and mentioned that the unique immunopreservation of neuropeptides has already been observed, and that we wanted to exploit this property in order to directly map neuromodulators onto connectomes.

*F) The Introduction does not touch at all on the nuchal system and the PDF-expressing neurons. What was the point of studying them? Why were they chosen as a case study?*

The nuchal organs represent chemosensory organs in annelids that have been extensively studied by annelid morphologists and are of considerable interest for marine sensory biology and for the comparative analysis of chemosensory circuits. In our whole-body siGOLD-labeled dataset we noticed that the sensory neurons of the nuchal organs showed a striking labeling with the PDF antibody. PDF represents a conserved neuropeptide that has been extensively studied in *Drosophila*. We discuss in more detail the implications of the circuitry and compare it to the olfactory circuitry in *Drosophila*.

*G) In Figure 2, contrast seems optimized to detect immunogold particles. Processing with CLAHE would better reveal cytoplasmic membranes while still preserving the saliency of immunogold particles. What was the algorithm chosen for improving contrast?*

The contrast on the images in Figure 2 was only minimally adjusted. We did not use the CLAHE algorithm. We provided high-resolution raw images showing excellent membrane contrast and gold labeling (Figure 2—source data 1). These are the images as we acquire them at the electron microscope. When the tiles are stitched together to generate the final layer, the tile with the best contrast was chosen, and the contrast was applied to all other tiles on the same layer using TrakEM2 (built-in commands such as: "Adjust images – Enhance contrast layer-wise" and "Adjust images – Set min and max layer-wise").

*H) In Figure 10, are the 1-synapse edges meaningful? What is the error rate and synapse count variance when redundantly or repeatedly mapping connections between neurons?*

We do not repeatedly trace the same neuron, instead we use the built-in functionality in Catmaid to review neuron traces by different users. Two expert neuroanatomists reviewed every neuron in the described circuit at least twice. Each synapse was visited at least four times, once from the presynaptic side and once from the postsynaptic site, by both tracers. All single-synapse connections have been confirmed. However, we do not know the functional significance of such singe-synapse connections. In the revised version we only show the main synaptic partners of the nuchal organs that receive many synaptic inputs (the IN^arc^ interneurons). We have now included a figure (Figure 10—figure supplement 2) showing all postsynaptic partners of the nuchal organ sensory neurons. It is clear that there are several other targets that receive five or less synaptic inputs from the nuchal organs, with 87 cells receiving only one synapse. These are unlikely to be all functionally relevant connections.

*I) The manuscript does not address the advantage over array tomography: far faster imaging and higher resolution, both in Z (thinner sections) and in XY. "With siGOLD we enjoy the full resolving power of the electron microscope" should mention transmission EM; array tomography uses SEM which offers far slower image acquisition.*

We think that the latest SEM technology can deliver imaging speeds that are comparable (and rapidly improving!) to TEM imaging. The imaging speed in TEM will also depend largely on the detection setup. We therefore cannot say which technology is faster. The sectioning thickness can also be varied in both approaches, and also does not represent a major difference. We discuss in the text that siGOLD could also be used in combination with SEM imaging.

*J) The Discussion does not address the issue of what has been learned by performing these labelings in* Platynereis *beyond merely proposing a circuit for chemotaxis.*

We have now extended the Discussion of the nuchal organ circuitry and compare the lateralization of synapse distribution to the olfactory circuitry in *Drosophila* and the visual circuitry in *Platynereis*.